# Hot Deformation Behavior and Microstructure Evolution of a Novel Al-Zn-Mg-Li-Cu Alloy

**DOI:** 10.3390/ma15196769

**Published:** 2022-09-29

**Authors:** Shuaishuai Wu, Baohong Zhu, Wei Jiang, Haochen Qiu, Yang Guo

**Affiliations:** 1GRIMAT Engineering Institute Co., Ltd., Beijing 101407, China; 2General Research Institute for Nonferrous Metals, Beijing 100088, China; 3GRINM Group Corporation Limited, Beijing 100088, China

**Keywords:** Al-Zn-Mg-Li-Cu alloy, hot compression, microstructure evolution, dynamic recrystallization, dynamic recrystallization mechanism

## Abstract

Lightweight structural alloys have broad application prospects in aerospace, energy, and transportation fields, and it is crucial to understand the hot deformation behavior of novel alloys for subsequent applications. The deformation behavior and microstructure evolution of a new Al-Zn-Mg-Li-Cu alloy was studied by hot compression experiments at temperatures ranging from 300 °C to 420 °C and strain rates ranging from 0.01 s^−1^ to 10 s^−1^. The as-cast Al-Zn-Mg-Li-Cu alloy is composed of an α-Al phase, an Al_2_Cu phase, a T phase, an η phase, and an η′ phase. The constitutive relationship between flow stress, temperature, and strain rate, represented by Zener–Hollomon parameters including Arrhenius terms, was established. Microstructure observations show that the grain size and the fraction of DRX increases with increasing deformation temperature. The grain size of DRX decreases with increasing strain rates, while the fraction of DRX first increases and then decreases. A certain amount of medium-angle grain boundaries (MAGBs) was present at both lower and higher deformation temperatures, suggesting the existence of continuous dynamic recrystallization (CDRX). The cumulative misorientation from intragranular to grain boundary proves that the CDRX mechanism of the alloy occurs through progressive subgrain rotation. This paper provides a basis for the deformation process of a new Al-Zn-Mg-Li-Cu alloy.

## 1. Introduction

With the rapid development of aerospace technologies, automobiles, and electronics, the demand for developing high-performance, low-density materials is increasing [1,2,3,4]. Al-Zn-Mg-Cu alloys are a promising engineering material, which have outstanding mechanical properties, low density, and excellent corrosion resistance [4,5,6]. To improve the comprehensive properties of alloys is the unremitting pursuit of researchers. The composition design optimization of Al-Zn-Mg-Cu alloys is an effective method to enhance their comprehensive properties [7,8,9,10]. Al-Zn-Mg-Cu alloys are a kind of age-strengthened aluminum alloy. As strengthening elements, Zn and Mg can greatly improve the mechanical properties of alloys by forming η′ and η phases [11]. However, the content of main strengthening elements in common aluminum alloys is limited (no more than 5 at% per element). Little attention has been paid to Al-Zn-Mg-Cu alloys with a high content of strengthening elements. The concept of entropy increasing in high-entropy alloys brings new inspiration to the design of aluminum alloys [12,13,14,15,16]. Researchers have sought to increase the contents of multiple alloying elements and at the same time strengthen and toughen the alloy. For example, Al_80_-Zn_5_-Mg_5_-Li_5_-Cu_5_ (at. %) alloys exhibit high tensile strength (674 MPa) and tenable ductility (7.5%) [14,17]. By further increasing the content of Zn element and slightly adjusting that of other elements, the compressive strength of Al_80_-Zn_14_-Mg_2_-Li_2_-Cu_2_ (at. %) alloys reaches 1000 MPa and maintains 20% plasticity [18].

Hot processing methods are a better choice for obtaining uniform microstructure and excellent mechanical properties in Al-Zn-Mg-Cu alloys [19]. In recent years, the optimization of hot processing methods has been the focus of research, with particular attention to the influence of the hot deformation temperature and strain rate on flow stress and microstructure. For instance, Zhou et al. [20] studied the effects of deformation temperature and strain rate on hot tensile deformation behaviors. Zhao et al. [4] investigated the dynamic recrystallization mechanism of an Al-Zn-Mg-Cu alloy at different strain rates and proposed two totally different constitutive equations. However, there are few studies on the hot deformation behavior of Al-Zn-Mg-Cu alloys with high alloying element content. Among them, Tang et al. [21] found that increases in Zn content mainly affect dynamic softening by promoting dynamic precipitation and inhibiting dynamic recovery and dynamic recrystallization. Al-Zn-Mg-Cu alloys with high alloying elements exhibit excellent mechanical properties at room temperature, but their hot workability degrades [18,22,23]. Therefore, it is significant to investigate the hot deformation behavior and microstructure evolution of highly alloyed Al-Zn-Mg-Li-Cu alloys during hot deformation to optimize thermomechanical processing.

This paper aims to systematically characterize the hot deformation behavior of the highly alloyed Al_78_-Zn_13_-Mg_5_-Li_2_-Cu_2_ alloy (atomic percent, at. %), establishing the constitutive relation, analyzing the mechanism of microstructure evolution during the hot deformation process, and providing a theoretical basis for the deformation processing.

## 2. Materials and Methods

The raw materials used in the experiment were aluminum ingots (Trillion Metals Co., Ltd., Beijing, China) with a purity of 99.99 wt.%, zinc particles (Trillion Metals Co., Ltd., Beijing, China) with a purity of 99.9 wt.%, magnesium ingots (Trillion Metals Co., Ltd., Beijing, China) with a purity of 99.9 wt.%, lithium particles (Trillion Metals Co., Ltd., Beijing, China) with a purity of 99.9 wt.%, and copper particles (Trillion Metals Co., Ltd., Beijing, China) with a purity of 99.9 wt.%. The Al_78_-Zn_13_-Mg_5_-Li_2_-Cu_2_ alloy (simply expressed as the 13Zn alloy) ingot was obtained by droplet ejection and the extremely high cooling rate allows for a fine grain size and a substantially segregation-free alloy ingot [24,25,26,27]. The detailed ingot preparation process was shown in our previous work [28]. All samples used in the experiment were taken at positions equidistant from the center of the ingot. The microstructure and properties of the alloy were studied in the as-cast condition. The actual chemical composition of the studied alloy is shown in Table 1. The density of the as-cast alloy measured by the Archimedes drainage method is 2.86 g·cm^−3^. Figure 1 shows the DSC curve of the 13Zn alloy during the heating process. The curve shows that the studied alloy has two obvious endothermic peaks during the heating process. The onset melting temperature of the first endothermic peak is 467 °C and the peak melting temperature is 495 °C, which corresponds to the dissolution of the high-melting-point phase in the alloy. There is only one obvious endothermic peak between 500 °C and 700 °C, which indicates that the initial melting temperature of the 13Zn alloy is 545 °C and the peak melting temperature is 585 °C. In addition, there are two less obvious endothermic peaks at 199 °C and 395 °C in the DSC curve, which correspond to the dissolution of the low-melting-point phase.

The samples used for the compression test were all cylinders with a diameter of 10 mm and a height of 15 mm. The compression tests were conducted at room temperature (RT), 300 °C, 330 °C, 360 °C, 390 °C, and 420 °C with strain rates of 0.01 s^−1^, 0.1 s^−1^, 1 s^−1^, and 10 s^−1^ by Gleeble-3500. Both ends of the sample were padded with graphite sheets and coated with grease to reduce the effect of friction. The samples were heated to the deformation temperature at a heating rate of 10 °C/s and held for 60 s to maintain a stable and uniform temperature. The samples were deformed to 50% of the original height and immediately quenched with water to keep the deformation microstructure.

The as-cast samples for microstructure observation were ground and polished. The deformed samples for microstructure observation were sectioned along the centerline parallel to the compression axis. Then, the microstructure was observed by scanning electron microscopy (SEM, JSM-7900F, JEOL Ltd., Tokyo, Japan) equipped with energy dispersive spectroscopy (EDS) and electron backscatter diffraction (EBSD). The EBSD observation for deformed specimens was carried out with a step size of 0.2 µm, operating at 20 kV. The EBSD data was analyzed using TSL OIM software (v7.0, EDAX Inc., Mahwah, NJ, USA). X-ray diffraction (XRD) was used to identify the phases at 30 kV with Cu Kα radiation. Transmission electron microscopy (TEM) was further employed to characterize the finely dispersed precipitates. The samples for TEM were prepared by double-jet electropolishing. High-angle annular dark-field imaging (HAADF), element mapping, high-resolution TEM (HRTEM), and selected area diffraction patterns (SADPs) were obtained by using an FEI-Talos F200X microscope.

The grain orientation spread (GOS) maps were obtained from EBSD data to distinguish between the recrystallized grains and the deformed grains according to their GOS values. The structures with a GOS value of less than 2° were considered recrystallized structures, a GOS value between 2° and 5° was considered to represent substructures, and the structures with a GOS value of more than 5° were considered deformed structures [29,30,31]. Grain boundary misorientation angles less than 10° are defined as low-angle grain boundaries (LAGBs) and are marked with thin light-yellow lines. Grain boundary orientation angle differences between 10° and 15° are defined as medium-angle grain boundaries (MAGBs) and are marked with red lines. Grain boundary orientation angle differences >15° are defined as high-angle grain boundaries (HAGBs) and are marked with black lines.

## 3. Results and Discussion

### 3.1. Microstructure of the As-Cast 13Zn Alloy

The microstructure of the as-cast 13Zn alloy is shown in Figure 2. The average grain size (15.07 μm) of the alloy is very fine, as shown in Figure 2a,c. The microstructure of the studied alloy consists of the network, bulk, long needle-like, and near-spherical phases. The network phases are distributed along the interface of the near-spherical phase, while the long needle-like phase is distributed within the grains of the near-spherical phase. Table 2 shows the EDS composition analysis of the studied alloy. In Figure 2b, P1, P2, and P3 are composed of the near-spherical phase, bulk phase, and network phases, respectively. Li element cannot be analyzed due to the limitations of EDS analysis. The content of Al element in the nearly spherical phase is close to 90 at. %, which means that it is the α-Al phase. The bulk phase contains a high content of Al and Cu elements, and the atomic ratio is close to 2:1. This suggests that the bulk phase may be the θ-Al_2_Cu phase. In addition to the higher contents of Al and Cu elements, the network phase also has higher contents of Zn elements and lower contents of Mg elements, which implies a combination of the θ-Al_2_Cu phase and the T phase (AlZnMgCu phase) [32,33,34]. Comparing the EBSD image of the as-cast alloy in Figure 2c,d, the nearly spherical phase is the α-Al phase and part of the network phase has a crystal structure consistent with the T-Mg_2_Zn_3_Al_4_ phase. This result is consistent with the EDS analysis in Table 2. As the long needle-like phase is too small, the phase distribution map by EBSD cannot be calibrated. By comparing the phases commonly found in Al-Zn-Mg alloys, this long needle phase could be the η-MgZn_2_ phase [35]. Figure 3 shows the XRD pattern of the as-cast alloy. The existence of the T phase, the Al_2_Cu phase, and the MgZn_2_ phase are clearly shown.

To further gain more information about the needle phase, a typical TEM image of the 13Zn alloy was shown in Figure 4. It can be seen that in addition to the long needle-like phase, there are also many plate-shaped nanoprecipitate phases in Figure 4a. By analyzing selected area diffraction patterns on the plate-shaped phases from zone Ⅰ, the results show that the weak diffraction spots located at the 1/3 $\left(02\bar{2}\right)$ Al and 2/3 $\left(02\bar{2}\right)$ Al indicate that these precipitates are η′ phases. The results are consistent with Ref [36,37]. Figure 4b presents the element mapping of precipitates in zone Ⅱ, the long needle-like phase and the plate-shaped phase, which are rich in Zn, Mg, and Cu elements, but depleted in Al element. It should also be noted that both grow laterally on {111} Al [38]. Thus, the long needle-like phase and the plate-shaped phase are the η phase and the η′ phase, respectively. The reason for the inclusion of the Cu element is that part of the Cu element is dissolved in the phase precipitation process [34]. From Figure 4c,d, the interplanar spacing of the (111) plane for the α-Al is 0.2357 nm, which is close to the value of 0.234 nm of the pure Al [39]. The reason for this difference may be the result of lattice distortion of the complex composition. Some of the η′ phases are only 2.13 nm wide, with better coherence with the α-Al phase. Taken altogether, the phase types in the alloy are the η′, η, Al_2_Cu, T, and α-Al phases. Therefore, the endothermic peaks at 199 °C and 395 °C of the DSC curve in Figure 1 correspond to the dissolution of the η′ and η phases, respectively [40]. It is reported that the onset temperature of the T phase in Al-Zn-Mg-Cu alloys is 479.5 °C and the dissolution temperature of the Al_2_Cu phase in Al-Cu alloys is between 460 °C and 530 °C [40,41]. This indicates that the peak dissolution temperature of 495 °C in Figure 1 is the dissolution of the T phase and the Al_2_Cu phase at the same time.

### 3.2. Flow Stress Behavior

Figure 5 shows the true compressive stress–strain curves of the 13Zn alloy at room temperature (RT) and at 300–420 °C with strain rates from 0.01 s^−1^ to 10 s^−1^. It was found that under different deformation conditions, the flow stress increased significantly with increases in strain at the initial stage due to the rapid increase and accumulation of dislocations. At RT, the alloy exhibits poor plastic deformation ability, and the specimen cracks rapidly when the true strain increases to 0.09–0.16. The larger true strain when the alloy cracked at high strain rates may be due to the adiabatic temperature rise. As the strain continued to increase at a deformation temperature of 300–420 °C, the flow stress gradually peaked, then reached a steady state or gradually decreased. This indicates that dynamic recovery (DRV) and dynamic recrystallization (DRX) occur during hot deformation. In addition, the flow stress is significantly influenced by the deformation temperature and strain rate. The flow stress decreases significantly with increases in the deformation temperature due to the softening effect at higher temperatures. At a strain rate of 10 s^−1^, the peak stress of the 13Zn alloy is 213.5 MPa at 300 °C, while it is reduced to 125.5 MPa at 420 °C, as shown in Figure 5a,e. The high strain rate shortens the deformation time and reduces the dynamic softening time. Therefore, the flow stress increases significantly with increases in the strain rate at the same deformation temperature. The typical features of the 13Zn alloy’s high-temperature true compressive stress–strain curves are consistent with those reported for a Al-Zn-Mg-Cu alloy [10,42].

### 3.3. Constitutive Equation

The peak stress of the material is an important parameter during hot deformation processing [19,43], and accurate prediction of peak stress is of great importance for deformation processing. By selecting the peak stress–stain data in Figure 5, we established the constitutive equation between peak stress, strain rate, and deformation temperature during hot compression. The interaction relationship between the flow stress σ, the deformation temperature *T*, and the strain rate $\dot{\varepsilon }$ of the alloy during hot deformation can usually be expressed as [44,45,46,47,48]:(1)$$Z=\dot{\varepsilon }\exp(Q/RT)=A{\left[sinh(\alpha \sigma )\right]}^{n}$$
where *Z* is the Zener–Hollomon parameter, $\dot{\varepsilon }$ is the deformation rate (s^−1^), *Q* is the deformation activation energy of the alloy (kJ·mol^−1^), *R* is the gas constant (the value is 8.314 J·mol^−1^·K^−1^), σ is the true stress (MPa), *T* is the deformation temperature (K), and *A*, *α*, and *n* are temperature-independent constants.

Equation (2) is based on Taylor expansion from Equation (1):(2)$$Z=\dot{\varepsilon }\exp\left(Q/RT\right)=\left\{\begin{array}{l} A_{1}\sigma^{n_{1}},\alpha \sigma <0.8\\ A_{2}\exp\left(\beta \sigma \right),\alpha \sigma >1.2 \end{array}\right.$$
where *α* = *β*/*n*_1_, *A*_1,_ and *A*_2_ are temperature-independent constants.

The following Equations (3) and (4) can be obtained by transforming Equations (1) and (2):(3)$$lnZ=ln\dot{\varepsilon }+Q/RT=lnA+nln[sinh(\alpha \sigma )]$$
(4)$$lnZ=ln\dot{\varepsilon }+Q/RT=\left\{\begin{array}{l} lnA_{1}+n_{1}ln\sigma ,\alpha \sigma <0.8\\ lnA_{2}+\beta \sigma ,\alpha \sigma >1.2 \end{array}\right.$$

By taking the derivative of 1/*T* on both sides of Equation (3), it can be derived that the values of *Q* are:(5)$$Q=R{\left.\frac{\partial ln\dot{\varepsilon }}{\partial ln[sinh(\alpha \sigma )]}\right|}_{T}•{\left.\frac{\partial ln[sinh(\alpha \sigma )]}{\partial (1/T)}\right|}_{\dot{\varepsilon }}$$

Figure 6 and Figure 7a exhibit the linear relationships of *ln*
$\dot{\varepsilon }\;$ versus *lnσ*, *ln*
$\dot{\varepsilon }$ versus *σ*, *ln*
$\dot{\varepsilon }$ versus *ln*[*sinh(ασ)*], *ln*[*sinh*(*ασ)*] versus 1000/*T*, and *lnZ* versus *ln*[*sinh(ασ)*]. Through linear regression analysis of the experimental data, the material constants *n* = 6.0563, *α* = 0.00916, and *lnA* = 31.60, and the deformation activation energy *Q* = 177.3 kJ·mol^−1^ of the 13Zn alloy can be obtained.

Finally, the constitutive equation of the peak stress of the 13Zn alloy can be obtained as:(6)lnZ=31.60+6.0563ln[sinh(0.00916σ)]

It can also be expressed as:(7)$$\dot{\varepsilon }=5.299\times 10^{13}{\left[sinh(0.00916\sigma )\right]}^{6.0563}\exp(-\frac{177.292\times 10^{3}}{RT})$$

Figure 7b presents the comparison of the experimental and calculated peak stress values of the 13Zn alloy at different temperatures and strain rates. The calculated value of the peak stress is quite close to the experimental value, and the correlation coefficient R^2^ is equal to 0.9847. This shows that the constitutive model of peak stress of the 13Zn alloy has good applicability.

It is well known that the magnitude of the activation energy *Q* reflects the degree of deformation difficulty for plasticity deformation and the thermodynamic mechanism of dislocation movement [21,49]. The activation energy of deformation *Q* for the studied alloy is 177.3 kJ·mol^−1^ at a strain of 0.69, which is higher than that of common Al-Zn-Mg-Cu alloys [21,50]. The self-diffusion activation energies for Al, Zn, Mg, Li, and Cu are 142, 102, 134, 55, and 197 kJ·mol^−1^, respectively [51]. Therefore, the self-diffusion activation energy value obtained by weighted mean according to its content [49] is 137.3 kJ·mol^−1^. The experimental activation energy is higher than the weighted mean self-diffusion activation energy and the self-diffusion activation energy for Al. The high density of η′ precipitates increases the deformation activation energy of the alloy by hindering the proliferation and movement of dislocations during deformation [50].

### 3.4. Microstructure Evolution

#### 3.4.1. The Effect of Deformation Temperatures on Microstructure

Figure 8 shows the XRD patterns of the 13Zn alloy after hot deformation at deformation temperatures of 300–420 °C. The results show that after deformation, the studied alloy is composed of an α (Al) phase, a T phase, a MgZn_2_ phase, and an Al_2_Cu phase. Compared with the as-cast alloy, the phase type of the hot deformed alloy did not change. According to the DSC curve in Figure 1, the T phase and the Al_2_Cu phase do not dissolve during thermal deformation, so they are always present after deformation. The existence of the MgZn_2_ phase may be precipitated by room temperature aging after dissolution.

Figure 9 shows the GOS maps of the 13Zn alloy at different deformation temperatures with a deformation rate of $\dot{\varepsilon }$ = 0.1 s^−1^ and *ε* = 0.69, a mixture of recrystallized grains represented in blue, the substructure represented in yellow, and deformed grains represented in red. It was found that the T phase (marked in black) always exists in the alloy at deformation temperatures ranging from 300 °C to 420 °C. A small amount of the bulk Al_2_Cu phase was not directly observed. Unlike in the as-cast structure, the T phase with a network distribution changes from a continuous distribution at grain boundaries to a discontinuous distribution after the hot deformation process. It has been reported that coarse intergranular distribution phases are easily broken by shear stress during hot compression deformation [52]. The fragmentation of the T phase may also be caused by shear deformation. At the same time, it is noted that the matrix α-Al phase is obviously recrystallized. The low deformation temperature makes DRV and DRX not enough, and therefore the alloys are dominated by deformed grains and subgrains with a few DRX grains (shown in Figure 9a). With increases in the deformation temperature, DRX grains increase rapidly, while the deformed grains decrease rapidly and disappear completely at 420 °C. More detailed recrystallized volume fractions and recrystallized grain size data are shown in Figure 10. At deformation temperatures ranging from 300 °C to 420 °C, the DRX volume fraction increased from 44.7% to 72.8%, and the recrystallized volume fraction increased rapidly at higher deformation temperatures. The higher increase in the recrystallized volume fraction at high deformation temperatures is due to the rapid migration of grain boundaries, allowing the DRX to progress sufficiently. Meanwhile, the average DRX grain size grew significantly with increases in the deformation temperature. The DRX grain size is only 2 μm at a deformation temperature of 300 °C. When the deformation temperature is 420 °C, the average DRX grain size (7.60 μm) is nearly half of the as-cast grain size (15.07 μm). The dispersed long needle-like η phase in the 13Zn alloy gradually dissolves with increases in the deformation temperature. The long needle-like dispersed precipitates are reported to stabilize the recrystallized grain size [53]. Therefore, as the deformation temperature exceeds the temperature at which the η phase begins to dissolve, the DRX grains grow significantly.

#### 3.4.2. Effect of Strain Rate on Microstructure

The GOS maps of the 13Zn alloy at different strain rates at a deformation temperature of 330 °C and *ε* = 0.69 are shown in Figure 11. The DRX volume fraction and recrystallized grain size data are shown in Figure 12. The microstructure after the hot compression deformation of the studied alloy is closely related to the strain rate. At a strain rate between 0.01 s^−1^ and 10 s^−1^, the studied alloy contains DRX grains and deformed grains. With increasing strain rate, the DRX grain size decreased rapidly, while the DRX volume fraction showed a trend of first increasing and then decreasing. When the strain rate $\dot{\varepsilon }$ = 0.01 s^−1^, the 13Zn alloy has sufficient time for nucleation and dynamic recrystallization growth, forming a microstructure with both coarse recrystallized grains and deformed grains. When the strain rate reaches 0.1 s^−1^, the grain boundaries do not have enough time to migrate, resulting in the recrystallized grains arising too late to grow sufficiently, thereby significantly refining the DRX grain size. By further increasing the strain rate to 1 s^−1^, the DRX is further increased, and the DRX grain size is further refined. The enhanced recrystallization is attributable to the high strain rate, which favors the formation of a large number of dislocations and structural defects in the 13Zn alloy, thereby increasing the driving force for DRX. When the strain rate is further increased to 10 s^−1^, the extremely short deformation time makes the migration of grain boundaries limited by time, and it is too late for the DRX to develop, resulting in uneven recrystallization.

### 3.5. DRX Mechanism of the 13Zn Alloy

The DRX mechanism is closely related to the misorientation angle [48,54]. Figure 13 presents the distribution of grain boundary misorientation angles at different deformation temperatures with a strain rate of 0.1 s^−1^. Figure 13a–c show that there is a large number of LAGBs in the microstructure of the 13Zn alloy after deformation below 360 °C. This indicates that a high density of dislocations still exists in the 13Zn alloy at lower deformation temperatures due to work hardening. At higher deformation temperatures, the microstructure contains only a small amount of small-angle grain boundaries. The lower dislocation density indicates an enhanced high-temperature softening effect. With increases in the deformation temperature, the number fraction of LAGBs gradually decreases while the number fraction of MAGBs and HAGBs gradually increases, as shown in Figure 13f. This is because the higher deformation temperature accelerates dislocation annihilation and rearrangement, which promotes the transition from LAGBs to HAGBs through MAGBs. It has been reported that the presence of MAGBs is necessary for the occurrence of subgrain rotation, which is an important feature for the occurrence of CDRX [55,56]. Subgrain boundaries gradually form LAGBs by continuously absorbing dislocations, and finally, new grains are formed. This is also consistent with the fact that Al with high stacking fault energy is prone to CDRX during hot deformation [57,58].

To further analyze the DRX mechanism of the 13Zn alloy, the point-to-point orientation difference and the point-to-origin cumulative orientation difference were obtained. Figure 14 shows OIM micrographs and orientation analysis of the 13Zn alloy deformed to a 0.69 true strain at different temperatures with a strain rate of 0.1 s^−1^. Note that the transformation from LAGBs to HAGBs was found at both 300 °C and 420 °C (thin light-yellow line to black line). The LAGBs formed through the DRV process promoted by dislocation arrangements continue to increase their misorientation by the continuing absorption of dislocation until new HAGBs are formed [46]. The process of forming new grains without pre-existing HAGBs suggests the occurrence of CDRX. Figure 14b–c,e–f depict that some point-to-point misorientation angles exceed 5° or even 10° at the grain interior, and the cumulative misorientation angles easily exceed 10° both at low and high deformation temperatures, which indicates that progressive subgrain rotation occurs within deformed grains [50]. The cumulative misorientation angler steps of 10°–15° indicate the presence of subgrain in CDRX. These salient features show that the mechanism of DRX for the 13Zn alloy during hot deformation can be attributed to the CDRX through progressive lattice rotation.

## 4. Conclusions

In this work, the deformation behavior and microstructure evolution of the Al_78_-Zn_13_-Mg_5_-Li_2_-Cu_2_ alloy was studied at temperatures ranging from 300 to 420 °C, and with the strain rate ranging from 0.01 s^−1^ to 10 s^−1^. The following conclusions can be drawn:The as-cast microstructure of the Al_78_-Zn_13_-Mg_5_-Li_2_-Cu_2_ alloy is composed of a near-spherical α-Al phase, a bulk Al_2_Cu phase, a network T phase, a long needle-like η phase, and a plate-like η′ phase. The phase type of the alloy did not change after hot deformation at 300–420 °C.The established constitutive equation can well predict the peak stress of the alloy, and the alloy shows a high deformation activation energy, due to the high density of η′ precipitates.Due to the disappearance of the dissolution and pinning effect of the η phase at high temperature, the average increase in the DRX grain size significantly increases from 2 μm at 300 °C to 7.6 μm at 420 °C. The DRX grain size decreases with increases in the strain rates at a deformation temperature of 330 °C. The fraction of DRX increases with increasing deformation temperature, while it first increases and then decreases with increasing strain rate.With increases in the deformation temperature, the transition rate from LAGBs to HAGBs through MAGBs increases. The DRX mechanism for the alloy during hot deformation is the continuous dynamic recrystallization (CDRX) through progressive lattice rotation mechanism.

## Figures and Tables

**Figure 1 materials-15-06769-f001:**
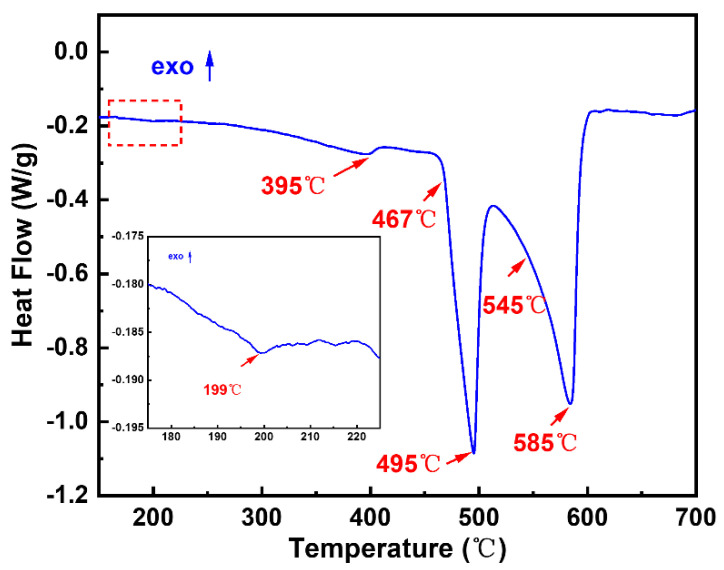
DSC curve of the as-cast 13Zn alloy.

**Figure 2 materials-15-06769-f002:**
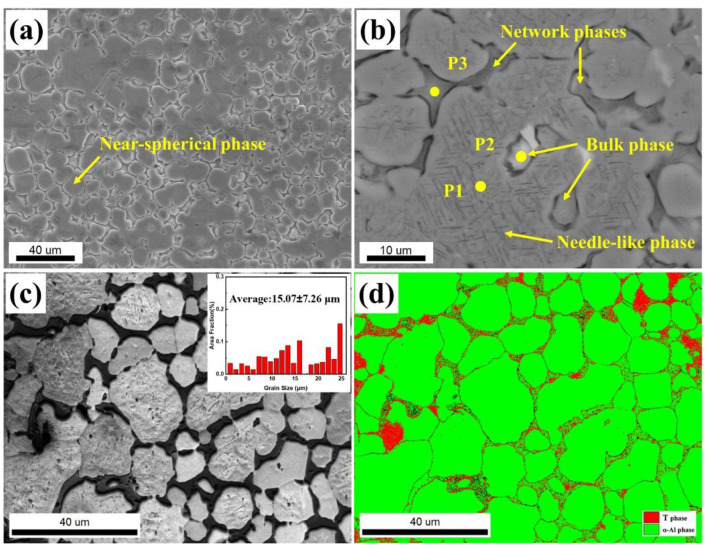
The as-cast microstructure of the 13Zn alloy viewed with (**a**) Low-magnification SEM; (**b**) High-magnification SEM; (**c**) IQ (Image Quality) image of EBSD; (**d**) Phase distribution image of EBSD.

**Figure 3 materials-15-06769-f003:**
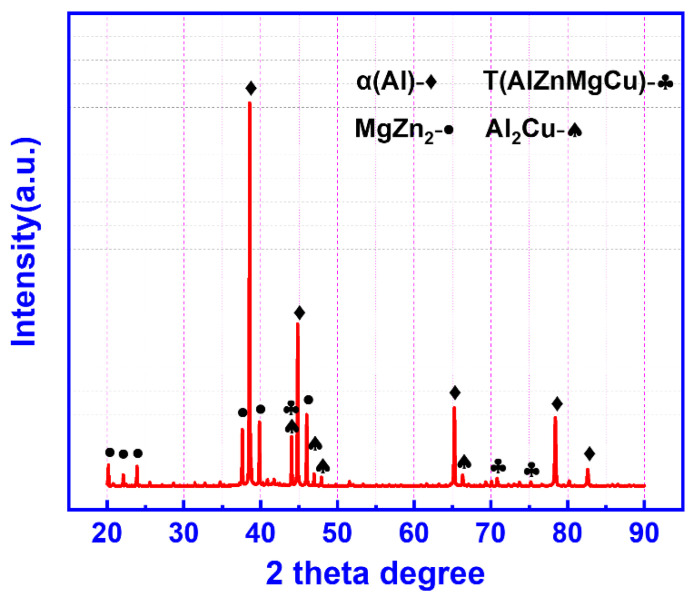
XRD patterns of the as-cast 13Zn alloy.

**Figure 4 materials-15-06769-f004:**
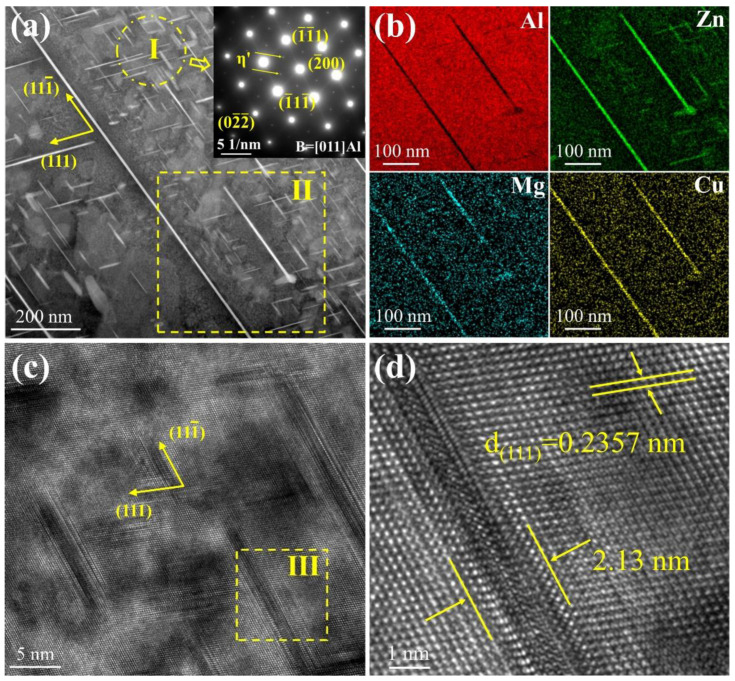
Typical TEM image of the 13Zn alloy (**a**) precipitate’s topography and SADPs along the [011] Al axis; (**b**) Element mapping of precipitates; (**c**) HRTEM image of η’ phase; (**d**) Zone III in image (**c**).

**Figure 5 materials-15-06769-f005:**
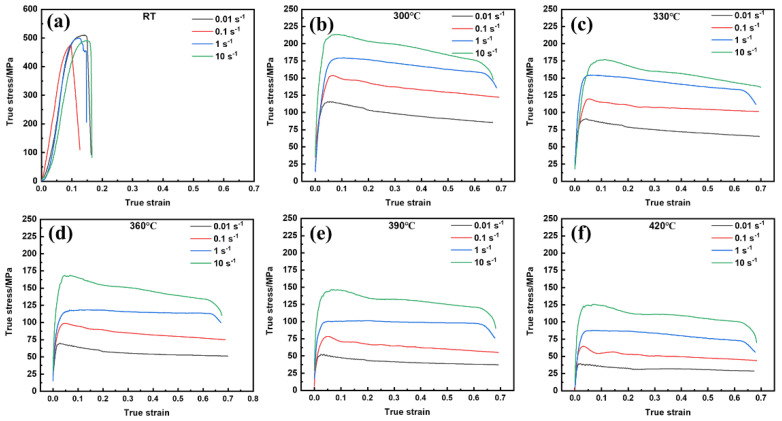
The true compressive stress–strain curves for the 13Zn alloy with strain rates from 0.01 s^−1^–10 s^−1^ at different temperatures (**a**) RT; (**b**) 300 °C; (**c**) 330 °C; (**d**) 360 °C; (**e**) 390 °C; (**f**) 420 °C.

**Figure 6 materials-15-06769-f006:**
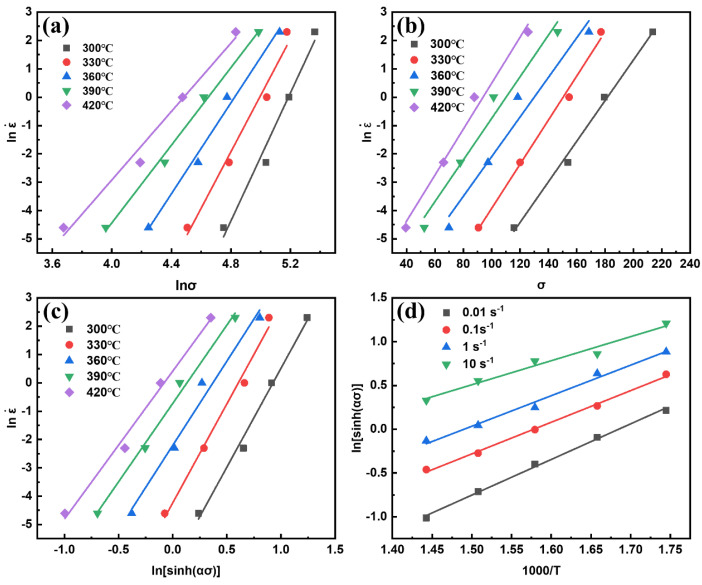
The linear relationships for different parameters of the studied alloy (**a**) *ln*$\dot{\varepsilon }\;$ versus *lnσ*; (**b**) *ln*$\dot{\varepsilon }$ versus *σ*; (**c**) *ln*$\dot{\varepsilon }$ versus *ln*[*sinh(ασ)*]; (**d**) *ln*[*sinh(ασ)*] versus 1000/*T*.

**Figure 7 materials-15-06769-f007:**
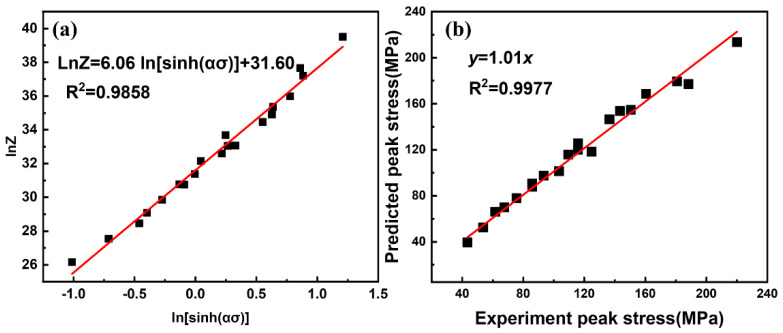
(**a**) The linear relationships of *lnZ* versus *ln*[*sinh(ασ)*]; (**b**) comparison of experimental and calculated peak stress of the 13Zn alloy.

**Figure 8 materials-15-06769-f008:**
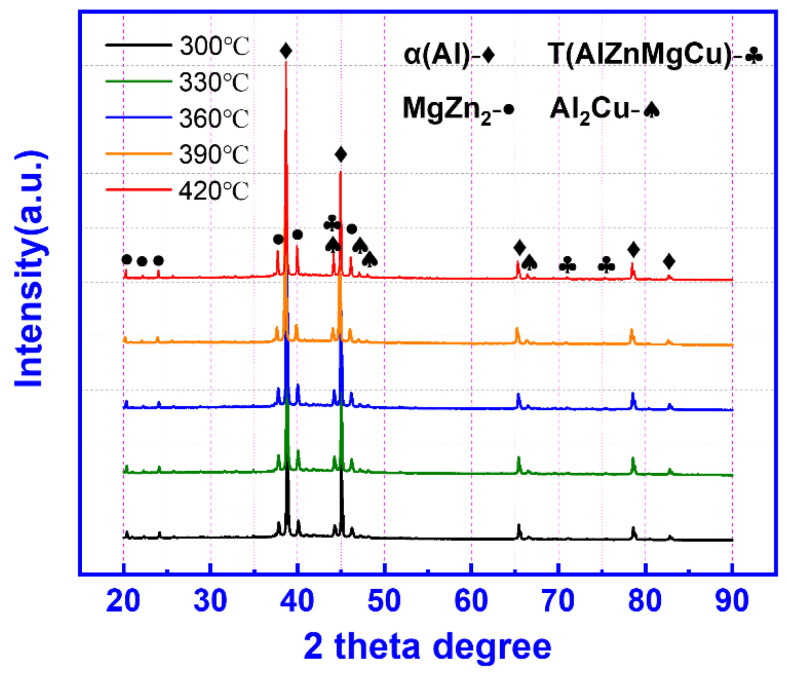
XRD patterns of the 13Zn alloy after hot deformation.

**Figure 9 materials-15-06769-f009:**
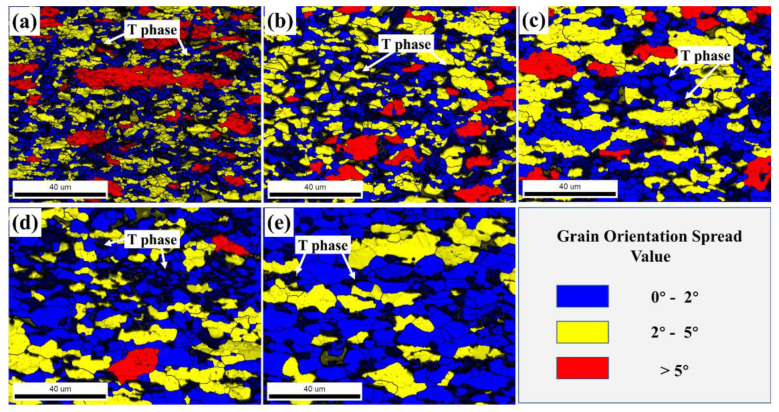
The GOS maps of the 13Zn alloy deformed to a 0.69 true strain at different temperatures with a strain rate of 0.1 s^−1^: (**a**) 300 °C; (**b**) 330 °C; (**c**) 360 °C; (**d**) 390 °C; (**e**) 420 °C.

**Figure 10 materials-15-06769-f010:**
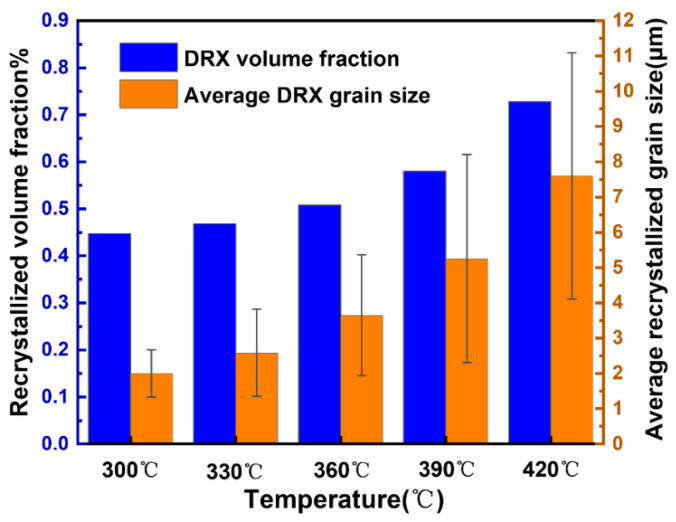
Recrystallized volume fraction and recrystallized grain size at different deformation temperatures.

**Figure 11 materials-15-06769-f011:**
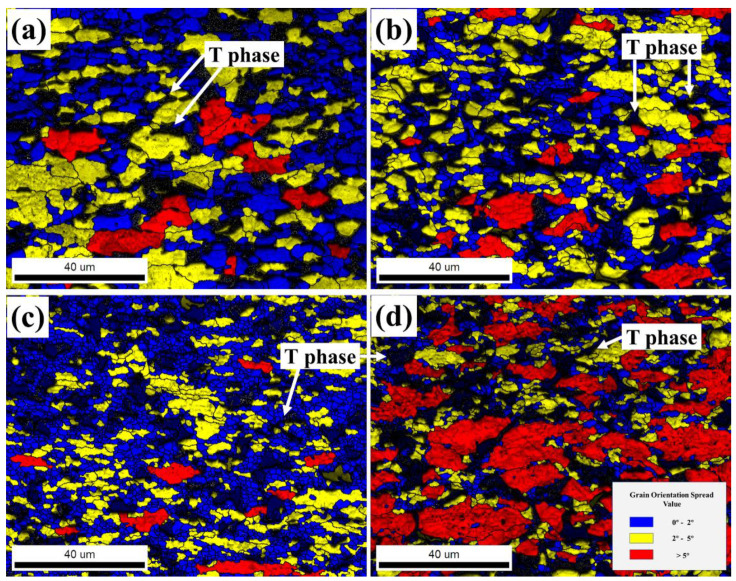
The GOS maps of the 13Zn alloy deformed to a 0.69 true strain at 330 °C with different strain rates: (**a**) 0.01 s^−1^; (**b**) 0.1 s^−1^; (**c**) 1 s^−1^; (**d**) 10 s^−1^.

**Figure 12 materials-15-06769-f012:**
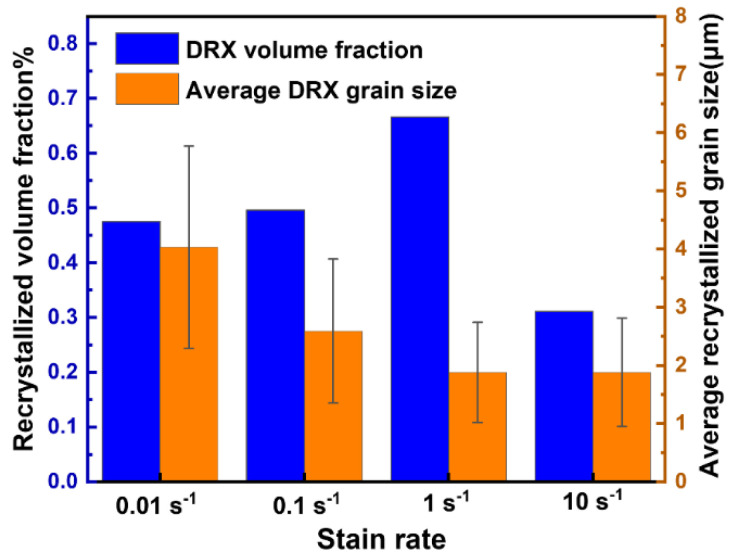
Recrystallized volume fraction and recrystallized grain size at different strain rates.

**Figure 13 materials-15-06769-f013:**
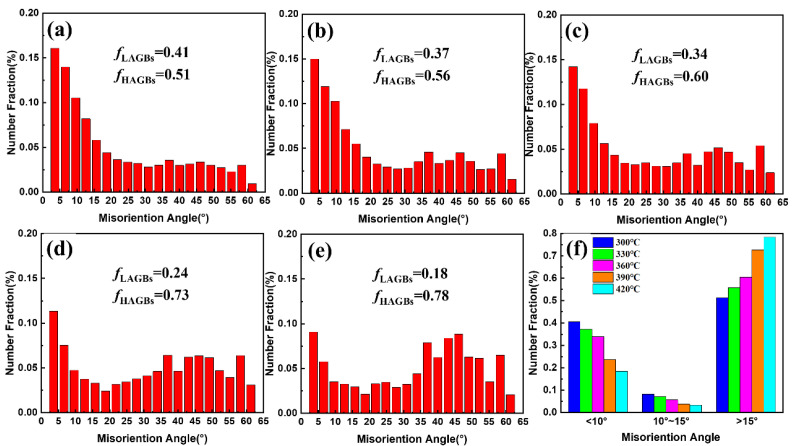
Distribution of grain boundary misorientation angles at different deformation temperatures with a strain rate of 0.1 s^−1^: (**a**) 300 °C; (**b**) 330 °C; (**c**) 360 °C; (**d**) 390 °C; (**e**) 420 °C; (**f**) Comparison of misorientation angles.

**Figure 14 materials-15-06769-f014:**
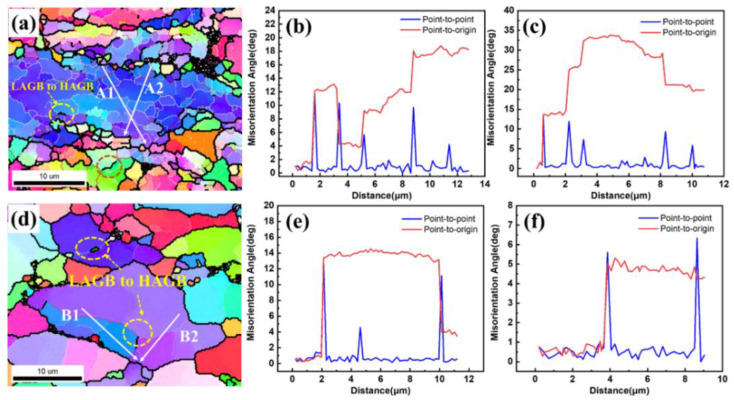
OIM micrographs and orientation analysis of the 13Zn alloy deformed to a 0.69 true strain at different temperatures with a strain rate of 0.1 s^−1^: (**a**) Typical OIM micrograph at a deformed temperature of 300 °C, (**b**,**c**) misorientations measured along the lines A1 and A2 marked in (**a**); (**d**) Typical OIM micrograph at deformed temperature 420 °C, (**e**,**f**) misorientations measured along the lines B1 and B2 marked in (**d**).

**Table 1 materials-15-06769-t001:** Actual chemical composition of the studied alloy (wt.%).

Element	Zn	Mg	Li	Cu	Al
Content	24.84	3.75	0.41	3.78	Bal.

**Table 2 materials-15-06769-t002:** EDS composition analysis results of P1 to P3 in Figure 2b (at. %).

Position	Zn	Mg	Li	Cu	Al
P1	7.73	0.64	-	1.91	89.81
P2	5.78	0.66	-	25.14	68.42
P3	12.59	1.11	-	25.92	60.39

## Data Availability

All data are available from the corresponding author upon reasonable request.

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
