# Peer review of "Hot Deformation Behavior and Microstructure Evolution of a Novel Al-Zn-Mg-Li-Cu Alloy"

_materials, 2022, doi:10.3390/ma15196769_

Round 1

Reviewer 1 Report

The manuscript titled " Hot deformation behavior and microstructure evolution of an Al-Zn-Mg-Li-Cu lightweight entropy alloy" is written in a well-structured way. The author has attempted to reveal the high temperature mechanical properties of mentioned Alloy at different strain rates. However, it would be great if the author could address the following questions.

 AS authors know Al78-Zn13-Mg5-Li2-Cu2 is not a high entropy alloy, even not medium high entropy, so authors are suggested to re-write the introduction part without mention to the LWHEA.  And also change the manuscript title. And don’t mention the LWEA in the paper, it does not have any meaning.

Details about fabrication processed of Al78-Zn13-Mg5-Li2-Cu2 should be added to the methodology part.

Physical properties (melting temperature and density) of the fabricated sample should be added

Please explain about the testing temperature selection (why did authors select 300 °C, 330 °C, 360 °C, 390 °C, and 420 °C for to conduct the test)

Authors are strongly suggested to add the XRD analysis of the as cast sample

EDX analysis have to add in result part

Stress -strain in room temperature should be added to the paper

Line 228 “T phase was broken during the hot deformation process” please explain about and discuss more about it

Line 239 ” Meanwhile, the DRX grain size has grown significantly with the increase in the deformation temperature” why this happened

Section 3.4.2. Why did authors discuss the effect of strain rate at 330 °C, NOT at 300 °C or 420 °C?

How does the phase interfaces play a role during high temperature deformation/DRX?

Author Response

Dear reviewer:

Thanks very much for your careful review and constructive suggestions with regard to our manuscript. We have made comprehensive and detailed changes according to your suggestions. Below are our itemized responses, with revisions in the revised manuscript.

  1. As authors know Al78-Zn13-Mg5-Li2-Cu2 is not a high entropy alloy, even not medium high entropy, so authors are suggested to re-write the introduction part without mention to the LWHEA. And also change the manuscript title. And don’t mention the LWEA in the paper, it does not have any meaning.

Respondence:

We totally accept your point of view on the concept of high entropy alloys. We have rewritten the introduction part and revised the manuscript about "LWEA". And changed the title of the manuscript to "Hot deformation behavior and microstructure evolution of a novel Al-Zn-Mg-Li-Cu alloy". 

  1. Details about fabrication processed of Al78-Zn13-Mg5-Li2-Cu2 should be added to the methodology part.

Respondence:

The detailed preparation steps of the ingot are mentioned in another article[1], which is " The process is as follows: the chamber is prefilled with nitrogen, raw materials were induction melted, and atomised with nitrogen at a gas pressure of 0.7 MPa. Driven by gas pressure, the atomised droplets were subsequently deposited onto a substrate. The distance of the atomising deposition was kept constant at 430 mm and the deposition time is about 13 min.". This is mentioned in the revised manuscript. 

  1. Physical properties (melting temperature and density) of the fabricated sample should be added

Respondence:

Thank you for your valuable and thoughtful comments. We obtained the melting point of the alloy by DSC curve (Figure 1), which is 585 °C. The sample density (2.86 g·cm-3) was measured by the Archimedes drainage method. Relevant information has been added to Section "2. Materials and Methods". 

  1. Please explain about the testing temperature selection (why did authors select 300 °C, 330 °C, 360 °C, 390 °C, and 420 °C for to conduct the test)

Respondence:

The hot deformation temperature of common aluminum alloys is 300 °C ~450 °C [2-4]. Considering that the content of Zn in the alloy is relatively high, the melting point of the alloy decreases. Therefore, in this experiment, the maximum deformation temperature is 420 °C and the minimum deformation temperature is 300 °C, and a certain temperature interval (30 °C) is set. 

  1. Authors are strongly suggested to add the XRD analysis of the as cast sample

Respondence:

We appreciate it very much for this good suggestion, and we have done it according to your ideas. The XRD analysis of the as-cast sample shows the existence of the α-Al phase, T phase, Al2Cu phase, and MgZn2 phase. 

  1. EDX analysis have to add in result part

Respondence:

We totally accept this suggestion. EDX analysis have been added in "3.1. Microstructures at as-cast state". 

  1. Stress-strain in room temperature should be added to the paper

Respondence:

Thanks a lot for your suggestion. We have supplemented the room temperature stress-strain curve of the Al-Zn-Mg-Li-Cu alloy in Figure 4. 

  1. Line 228 “T phase was broken during the hot deformation process” please explain about and discuss more about it.

Respondence:

The initial T phase with a network distribution changes from continuous distribution at grain boundaries to discontinuous distribution after thermal compression deformation. It has been reported that coarse intergranular distribution phases are easily broken by shear stress during hot compression deformation [5]. Therefore, the fragmentation of the T phase may also be caused by shear deformation.

  1. Line 239 “Meanwhile, the DRX grain size has grown significantly with the increase in the deformation temperature” why this happened.

Respondence:

Through the supplementary DSC curves and further analysis, the dispersed long needle-like η phase in the Al-Zn-Mg-Li-Cu alloy gradually dissolves with the increase of the deformation temperature. The long needle-like dispersed precipitates are reported to stabilize the recrystallized grain size [6]. Therefore, as the deformation temperature exceeds the temperature at which the η phase begins to dissolve, the dynamically recrystallized grains grow significantly. 

  1. Section 3.4.2. Why did authors discuss the effect of strain rate at 330 °C, NOT at 300 °C or 420 °C?

Respondence:

In fact, the effect of strain rate on microstructure evolution is consistent at all deformation temperatures. There is no difference between the strain rate on the microstructure evolution at the hot deformation temperature of 300/330/420℃. In order to make the article not lengthy, we choose one temperature to express. 

  1. How does the phase interfaces play a role during high temperature deformation/DRX?

Respondence:

The second phase often plays a non-negligible role in the recrystallization process. The finely dispersed precipitates hinder the boundary movement through the Zener drag effect, slowing down recrystallization and grain growth [7, 8]. Coarse second phase can accelerate recrystallization by particle stimulated nucleation (PSN) [9, 10]. In the newly supplemented DSC curve (Figure 1), it can be seen that the η' phase has been dissolved or transformed into a stable η phase between the deformation temperature of 300 °C and 420 °C. The phases existing in the alloy during hot deformation are long needle-like η phase, coarse T phase and Al2Cu phase (see Figure 8. XRD patterns of the 13Zn alloy after hot deformation.). Coarse T phase and Al2Cu phase are able to facilitate DRX by PSN. The long needle-like η phase is an incoherent interface with the matrix α-Al, which can inhibit recrystallization through the interaction of dislocations. And it has been reported that the long needle-like dispersed phase can strongly inhibit the DRX grain growth [6].

Reference

  1. Jiang, W.; Tao, S.; Qiu, H.; Wu, S.; Zhu, B., Precipitation transformation and strengthening mechanism of droplet ejection lightweight medium-entropy AlZnMgCuLi alloy. Journal of Alloys and Compounds 2022, 922, 166152.
  2. Guo, Y.; Zhang, J.; Zhao, H., Microstructure evolution and mechanical responses of Al–Zn–Mg–Cu alloys during hot deformation process. Journal of Materials Science 2021, 56, (24), 13429-13478.
  3. Zhao, J.; Deng, Y.; Xu, F.; Zhang, J., Effects of initial grain size of Al-Zn-Mg-Cu alloy on the recrystallization behavior and recrystallization mechanism in isothermal compression. Metals 2019, 9, (2), 110.
  4. Yang, Q.; Deng, Z.; Zhang, Z.; Liu, Q.; Jia, Z.; Huang, G., Effects of strain rate on flow stress behavior and dynamic recrystallization mechanism of Al-Zn-Mg-Cu aluminum alloy during hot deformation. Materials Science and Engineering: A 2016, 662, 204-213.
  5. Tang, Y.; Xiao, D.; Huang, L.; You, R.; Zhao, X.; Lin, N.; Ma, Y.; Liu, W., Dynamic microstructural evolution of Al-Cu-Li alloys during hot deformation. Materials Characterization 2022, 191, 112135.
  6. Chang, K.; Feng, W.; Chen, L.-Q., Effect of second-phase particle morphology on grain growth kinetics. Acta Materialia 2009, 57, (17), 5229-5236.
  7. Nes, E.; Ryum, N.; Hunderi, O., On the Zener. Acta Mater 1985, 33, 11-22.
  8. Huang, K.; Logé, R. E., A review of dynamic recrystallization phenomena in metallic materials. Materials & Design 2016, 111, 548-574.
  9. Adam, K. F.; Long, Z.; Field, D. P., Analysis of particle-stimulated nucleation (PSN)-dominated recrystallization for hot-rolled 7050 aluminum alloy. Metallurgical and Materials Transactions A 2017, 48, (4), 2062-2076.
  10. Zang, Q.; Chen, H.; Lee, Y.-S.; Yu, H.; Kim, M.-S.; Kim, H.-W., Improvement of anisotropic tensile properties of Al-7.9 Zn-2.7 Mg-2.0 Cu alloy sheets by particle stimulated nucleation. Journal of Alloys and Compounds 2020, 828, 154330.

Reviewer 2 Report

1.     The authors used different hot deformation temperatures and maximum is at 420 deg. C, which is nearly melting of Zn. Is the chemical composition of the hot deformed susbtrates available to confirm the loss of elements during the process?

2.     The criteria to deice the peak stress must be defined with a standard method.

3.     In figure 7, please add the statistical error data for the grain size and volume.

4.     The size of Fig 10 and 11 need to be increased to analyze them in detail.

5.     Please provide the phase analysis of both wrought and hot deformed alloys to compare.

6.     Is Fig 2 showing wrought alloy TEM analysis or hot f\deformed?, it should indicate the presence of thick linear lines in Fig2a.

7.     Please provide the linear regression equation for Fig. 5.

8.     Is the T phase is not available for 360 Deg. C condition, as per Fig 6 indications.

9.     What is the maximum grain orientation spread, and what should be added to fig 6.

10.  The conclusions should mention whether the increase of grain size and a fraction of DRX is the same at every strain rate other than 1 S-1.

Author Response

Dear reviewer:

Thanks very much for taking your time to review this manuscript. We really appreciate all your comments and suggestions. Below are our itemized responses, with revisions in the revised manuscript.

  1. The authors used different hot deformation temperatures and maximum is at 420 deg. C, which is nearly melting of Zn. Is the chemical composition of the hot deformed susbtrates available to confirm the loss of elements during the process?

Respondence:

Although the hot deformation temperature is close to the melting point of Zn element, we do not find Zn precipitation on the surface of the specimen after hot compression. No Zn phase was found in the XRD pattern of the alloy after hot deformation. There is no obvious exothermic peak found in the DSC curve of the alloy about 419 ℃ (The melting point of Zn), so we think there is no element loss during the hot deformation at 420 ℃. 

  1. The criteria to deice the peak stress must be defined with a standard method.

Respondence:

We totally agree with you. The selection of the peak stress in this experiment is the value at which the compressive stress in the compressive stress-strain curve rises to the maximum value and begins to decrease or stabilize. 

  1. In figure 7, please add the statistical error data for the grain size and volume.

Respondence:

Thank you very much for your valuable suggestion. The statistical error data for DRX grain size were added to Figure 10 (Original Figure 7) and Figure 12 (Original Figure 9). The recrystallization volume fraction is based on data from one region, so there is no statistical error. 

  1. The size of Fig 10 and 11 need to be increased to analyze them in detail.

Respondence:

We fully accept your suggestion. Relevant modifications have been added to the "3.5 DRX mechanism of 13Zn alloy" section.

  1. Please provide the phase analysis of both wrought and hot deformed alloys to compare.

Respondence:

Thank you very much for your valuable suggestion. We have supplemented the manuscript with XRD phase analysis as-cast and after thermal deformation at different temperatures.

  1. Is Fig 2 showing wrought alloy TEM analysis or hot deformed? it should indicate the presence of thick linear lines in Fig2a.

Respondence:

Figure 4 (Original Figure 2) shows the as-cast state of the alloy. As the "thick line" phase in Figure 4a is similar to the η' phase in element composition and morphology, and both grow laterally on {111} Al. These features suggest the thick line is the long needle-like η phase [1]. 

  1. Please provide the linear regression equation for Fig. 5.

Respondence:

Thank you very much for your valuable suggestion, the linear regression equation has been added and Figure 7 (Original Figure 5) has been revised in the revised manuscript. The linear regression equations of Figure 7a and Figure 7b are LnZ=6.06 ln[sinh(ασ)] +31.60 and y=1.01x, respectively.

  1. Is the T phase is not available for 360 Deg. C condition, as per Fig 6 indications.

Respondence:

The T phase always exists at the deformation temperature of 300~420 ℃, which is confirmed by the newly supplemented XRD pattern of the alloy after hot deformation. We are very sorry that we forgot to mark the T-phase in Figure 9 (Original Figure 6(c)). New images have been added in the revised manuscript.

  1. What is the maximum grain orientation spread, and what should be added to fig 6.

Respondence:

It is generally regarded that the structures with a GOS value of less than 2° were considered recrystallized structures, a GOS value between 2° and 5° was considered substructures, and the structures with a GOS value of more than 5° are considered deformed structures [2, 3]. The maximum GOS value of different deformed grains is different, so the maximum GOS value is not given, but it is clearly pointed out that about 5° are deformed grains. The GOS value range annotations in Figure 9 (Original Figure 6) and Figure 11 (Original Figure 8) have been revised in the revised manuscript.

 10. The conclusions should mention whether the increase of grain size and a fraction of DRX is the same at every strain rate other than 1 s-1.

Respondence:

We totally accept this suggestion. In fact, the increase of grain size and a fraction of DRX is the same at every strain rate. For convenience, we choose the strain rate of 1 s-1 for illustration.

Reference

  1. Li, Y.-Y.; Kovarik, L.; Phillips, P. J.; Hsu, Y.-F.; Wang, W.-H.; Mills, M. J., High-resolution characterization of the precipitation behavior of an Al–Zn–Mg–Cu alloy. Philosophical Magazine Letters 2012, 92, (4), 166-178.
  2. Ma, K.; Liu, Z.; Bi, S.; Zhang, X.; Xiao, B.; Ma, Z., Microstructure evolution and hot deformation behavior of carbon nanotube reinforced 2009Al composite with bimodal grain structure. Journal of Materials Science & Technology 2021, 70, 73-82.
  3. Mokdad, F.; Chen, D.; Liu, Z.; Ni, D.; Xiao, B.; Ma, Z., Three-dimensional processing maps and microstructural evolution of a CNT-reinforced Al-Cu-Mg nanocomposite. Materials Science and Engineering: A 2017, 702, 425-437.

Round 2

Reviewer 2 Report

Revised version is improved in all aspects.

Author Response

Dear reviewer:

Thanks very much again for taking the time to review this manuscript. We are really grateful for your valuable advice. We have carefully checked and improved the English language and style in the revised manuscript and have asked colleagues who are fluent in English to check the English writing. All changes are highlighted in red in the revised manuscript.

Yours sincerely,

Mr. Shuaishuai Wu
